# Mutations derived from horseshoe bat ACE2 orthologs enhance ACE2-Fc neutralization of SARS-CoV-2

Huihui Mou[1⊙], Brian D. Quinlan[1⊙], Haiyong Peng[1⊙], Guanqun Liu[2], Yan Guo[1], Shoujiao Peng[1], Lizhou Zhang[1], Meredith E. Davis-Gardner[1¤a¤b], Matthew R. Gardner[1¤b¤c], Gogce Crynen[3], Lindsey B. DeVaux[1], Zhi Xiang Voo[1], Charles C. Bailey[4], Michael D. Alpert[4], Christoph Rader[1]*, Michaela U. Gack[2]*, Hyeryun Choe[1]*, Michael Farzan[1]*

1 Department of Immunology and Microbiology, The Scripps Research Institute, Jupiter, FL, United States of America, 2 Florida Research and Innovation Center, Cleveland Clinic, Port Saint Lucie, FL, United States of America, 3 Bioinformatics and Statistics Core, The Scripps Research Institute, Jupiter, FL, United States of America, 4 Emmune, Inc., Juno Beach, FL, United States of America

⊙ These authors contributed equally to this work.
¤a Current address: Emory Vaccine Center, Emory University School of Medicine, Atlanta, Georgia, United States of America
¤b Current address: Yerkes National Primate Research Center, Emory University, Atlanta, Georgia, United States of America
¤c Current address: Division of Infectious Diseases, Department of Medicine, Emory University School of Medicine, Atlanta, Georgia, United States of America
* crader@scripps.edu (CR); gackm@ccf.org (MUG); hchoe@scripps.edu (HC); mfarzan@scripps.edu (MF)

**Data Availability Statement:** All relevant data are within the manuscript and its Supporting Information files.

## Abstract

The severe acute respiratory syndrome coronavirus 2 (SARS-CoV-2) spike (S) protein mediates infection of cells expressing angiotensin-converting enzyme 2 (ACE2). ACE2 is also the viral receptor of SARS-CoV (SARS-CoV-1), a related coronavirus that emerged in 2002–2003. Horseshoe bats (genus *Rhinolophus*) are presumed to be the original reservoir of both viruses, and a SARS-like coronavirus, RaTG13, closely related to SARS-CoV-2, has been identified in one horseshoe-bat species. Here we characterize the ability of the S-protein receptor-binding domains (RBDs) of SARS-CoV-1, SARS-CoV-2, pangolin coronavirus (PgCoV), RaTG13, and LyRa11, a bat virus similar to SARS-CoV-1, to bind a range of ACE2 orthologs. We observed that the PgCoV RBD bound human ACE2 at least as efficiently as the SARS-CoV-2 RBD, and that both RBDs bound pangolin ACE2 efficiently. We also observed a high level of variability in binding to closely related horseshoe-bat ACE2 orthologs consistent with the heterogeneity of their RBD-binding regions. However five consensus horseshoe-bat ACE2 residues enhanced ACE2 binding to the SARS-CoV-2 RBD and neutralization of SARS-CoV-2 pseudoviruses by an enzymatically inactive immunoadhesin form of human ACE2 (hACE2-NN-Fc). Two of these mutations impaired neutralization of SARS-CoV-1 pseudoviruses. An hACE2-NN-Fc variant bearing all five mutations neutralized both SARS-CoV-2 pseudovirus and infectious virus more efficiently than wild-type hACE2-NN-Fc. These data suggest that SARS-CoV-1 and -2 originate from distinct bat species, and identify a more potently neutralizing form of soluble ACE2.

**Funding:** This work was supported by a National Institutes of Health SARS-CoV-2 supplement to the award R01 AI129868 (MF). The funders had no role in study design, data collection and analysis, decision to publish, or preparation of the manuscript.

## Author summary

The severe acute respiratory syndrome coronavirus 2 (SARS-CoV-2), like the closely related virus SARS-CoV (SARS-CoV-1), infects cells by interacting with the cellular receptor angiotensin-converting enzyme 2 (ACE2). This interaction is mediated by the viral spike (S) protein through an independently folded subdomain, described as its receptor-binding domain (RBD). The susceptibility of a species to SARS-CoV-1 or -2 infection correlates with the binding affinity of their respective RBD for the ACE2 orthologs of that species. We therefore investigated the binding affinity of the RBD regions of multiple SARS-like coronaviruses with a range of ACE2 orthologs. Our results are consistent with the hypothesis that pangolins serve as an intermediate between humans and horseshoe bats. We further observed a high level of variability in the ability of the SARS-CoV-2 RBD to bind horseshoe bat ACE2 orthologs, suggesting ongoing selection pressure on their receptor ACE2 proteins from SARS-like viruses. Indeed, mutations derived from different horseshoe bat orthologs introduced into a soluble form of human ACE2 differentially impacted SARS-CoV-1 and SARS-CoV-2 S-protein-mediated infection. A combination of five residues present in multiple horseshoe bats increased the ability of a soluble form of ACE2 to neutralize SARS-CoV-2 S-protein-mediated infection. Thus horseshoe bats ACE2 orthologs can provide insight useful to improving the potency of ACE2-based therapeutics.

## Introduction

Coronaviruses are enveloped positive-strand RNA viruses of the family *Coronaviridae* [1]. At least seven coronaviruses infect humans. Four of these, HCoV-229E, -OC43, -HKU1, and -NL63 cause mild upper respiratory tract symptoms in most infected persons. In contrast MERS-CoV and SARS-CoV-1 cause severe, often fatal, infections. The recently emerged SARS-CoV-2 is closely related to SARS-CoV-1, and causes the sometimes fatal disease COVID-19.

SARS-CoV-2, like SARS-CoV-1, uses the receptor ACE2 to infect cells [2, 3]. Association with ACE2 and infection of ACE2-expressing cells is mediated by the SARS-CoV-1 and SARS-CoV-2 spike (S) proteins [4, 5]. These S proteins are type I viral glycoproteins similar to influenza hemagglutinin and the HIV-1 envelope glycoprotein [6]. Like these latter envelope proteins, the S protein is processed into two domains, S1 and S2 [4, 5], either in the virus-producing cell (SARS-CoV-2) or in the ACE2-expressing target cell (SARS-CoV-1). S1 binds ACE2, whereas S2 anchors the S protein to the viral membrane and mediates fusion with the target-cell membrane. Somewhat unusually for type I entry proteins, the S1 domains of SARS-CoV-1 and -2 include distinct, independently folded receptor-binding domains (RBDs) of approximately 200 amino acids [7–9]. The RBD is the primary neutralizing epitope on the SARS-CoV-2 S1 protein [10, 11].

In addition to these human viruses, a number of SARS-like viruses (subgenus *Sarbecovirus* of the *Betacoronaviruses* genus) have been identified in horseshoe bats (genus *Rhinolophus*) [12–14]. These bats are also the presumed long-term reservoirs of SARS-CoV-1 and -2, but it remains unclear exactly which *Rhinolophus* species serves as the most recent bat host of each virus. Although some SARS-like viruses have deletions in their RBDs, precluding their use of ACE2, other SARS-like viruses, like SARS-CoV-1 and -2, have intact RBDs and retain their ability to bind and enter cells through ACE2 [8, 12, 15, 16]. A sequence from one such SARS-

like virus, RaTG13, obtained from the horseshoe bat species *R. affinis* is closely related to SARS-CoV-2 (96.2% nucleotide identity) [17]. *R. affinis* is also the source of LyRa11, a SARS-like coronavirus more similar to SARS-CoV-1 [18].

Abundant data implicate the palm civet (*Paguma larvata*) as a reservoir intermediate of SARS-CoV-1 [15]. For example, the palm-civet ACE2 ortholog is an efficient receptor for SARS-CoV-1 [19]. Also, SARS-CoV-1 has been found in palm civets at an exotic animal market in Guangdong province, where the virus first infected humans in 2002 [20]. Finally, a second independent human transmission of SARS-CoV-1, which occurred in the winter of 2003, was directly traced to a restaurant serving palm civets [21]. Analogously, the pangolin has been suggested to be a reservoir intermediate for SARS-CoV-2, and a closely related SARS-like virus that has been predicted to use ACE2 has been detected in this species [22]. These pangolins showed signs of coronaviral diseases, more consistent with an intermediate host than a long-term reservoir. The close similarity of the RBD from this pangolin coronavirus and the SARS-CoV-2 RBD has suggested that SARS-CoV-2 arose from a recombination event between a pangolin- and a bat-derived coronavirus, although it remains undetermined in what species this putative recombination event occurred [22, 23].

Soluble forms of ACE2, including its immunoadhesin form ACE2-Fc (also described as ACE2-Ig), neutralize both SARS-CoV-1 and SARS-CoV-2 [3, 4, 24]. ACE2-Fc may be useful for treating infected persons, and, in the absence of an effective vaccine, it might protect individuals from an initial infection [25].

Here we characterize twelve orthologs of ACE2, including those of pangolin and five horseshoe bat species, for their ability to bind the RBDs of five SARS-like viruses: SARS-CoV-1, SARS-CoV-2, PgCoV, LyRa11, and RaTG13. We observed a high level of variation among five horseshoe-bat ACE2 orthologs in their ability to associate with these RBD, but that orthologs with horseshoe-bat consensus residues bound all RBDs efficiently. Introduction of five consensus horseshoe-bat residues into human ACE2 enhanced binding to the SARS2-RBD and neutralization of SARS-CoV-2 S-protein pseudotyped retroviruses (SARS2-S-PV) as well as infectious SARS-CoV-2. These data indicate that elements of its reservoir-species ACE2 ortholog can be used to improve the neutralization potency of ACE2-Fc.

## Results

### Characterization of binding of the RBD-Fc of five SARS-like viruses to cells expressing 12 ACE2 orthologs

ACE2 utilization has been predictive of the susceptibility of a species to SARS-CoV-1 infection and provided useful insight to the zoonotic origins of this virus [19, 26]. We therefore initiated similar studies of SARS-CoV-2. We first measured the ability of six RBD immunadhesins (RBD-Fc) to associate with twelve ACE2 orthologs expressed on HEK293T cells (**Figs 1 and S1A–S1C**). All five RBDs from SARS-like viruses, namely those of SARS-CoV-1 (SARS1-RBD) and SARS-CoV-2 (SARS2-RBD), as well as RaTG13-RBD, PgCoV-RBD, and LyRa11-RBD detectably bound cells expressing human ACE2, whereas the control RBD of MERS-CoV, which binds the dipeptidyl-peptidase IV (DPP4), did not. Notably, the PgCoV-RBD, which differs from SARS2-RBD by seven amino-acids (**S1A Fig**), bound human ACE2 similarly to or more efficiently than did SARS1-RBD or SARS2-RBD, whereas the RaTG13-RBD bound significantly less efficiently. Differences between SARS1-RBD and SARS2-RBD were not significant in this assay, likely because higher avidity binding of dimeric RBD-Fc to cell expressed ACE2. However surface-plasmon resonance studies (SPR; **Figs 2A and 2B and S2A and S2B**) detected a three-fold difference in affinity between SARS1-RBD and SARS2-RBD when monomeric ACE2 was evaluated for binding to immobilized RBD-Fc, consistent with previous

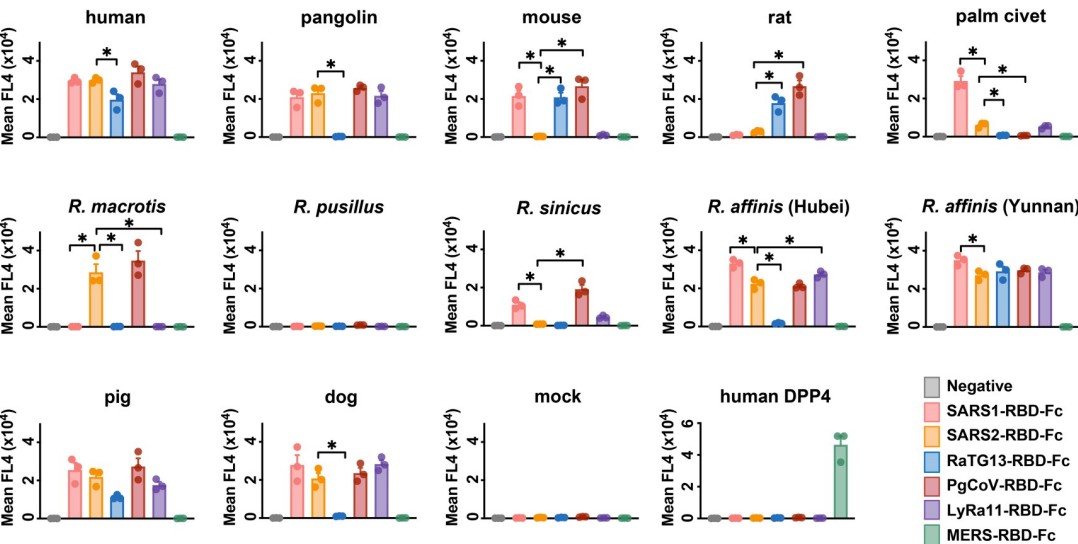

**Fig 1. The receptor-binding domains (RBD) of SARS-related coronaviruses S proteins bind differentially to ACE2 orthologs.** HEK239T cells transfected to express the ACE2 orthologs of the indicate species, human DPP4 (the receptor of MERS-CoV), or with vector alone (mock) were assayed by flow cytometry for their ability to bind SARS1-RBD-Fc, SARS2-RBD-Fc, RaTG13-RBD-Fc, PgCoV-RBD-Fc, LyRa11-RBD-Fc or MERS-RBD-Fc. Bars display the results of three independent experiments and the mean fluorescence intensity was normalized with relative ACE2 expressing levels shown in S1C Fig that determined by an antibody recognizing an amino-terminal myc-tag. One-way ANOVA was used to test difference across RBDs within each ACE2 ortholog and upon low $p$ value, Dunnett's multiple comparison test was used to compare each RBD with SARS2-RBD-Fc (*indicates $p < 0.05$).

reports [4]. These SPR studies also confirmed the higher affinity of the PgCoV-RBD for human ACE2 than either the SARS1- or SARS2-RBD observed in cell-binding studies. The ability of each of these RBD-Fc to inhibit SARS-CoV-1 and SARS-CoV-2 S-protein pseudo-virus (SARS1-S-PV, SARS2-S-PV, respectively) entry into human ACE2-expressing cells also correlated with their binding affinities for human ACE2 (**Fig 2C**), and dimeric ACE2-Fc neutralized SARS1- and SARS2-S-PV more efficiently than did monomeric ACE2-CT (**S2C and S2D Fig**).

We also characterized these RBD-Fc constructs to bind the ACE2 orthologs of a number of other species. Of note, PgCoV-RBD and RaTG13-RBD, both closely related to the SARS2-RBD, but not the SARS2-RBD itself, bound mouse and rat ACE2 efficiently (**Fig 1**). As previously reported, the SARS1-RBD also bound mouse but not rat ACE2 [19, 26]. Also interestingly, only SARS1-RBD bound the ACE2 ortholog of palm civets, a known reservoir intermediate of SARS-CoV-1 [19, 20, 22]. Even the LyRa11-RBD, which is very similar to the SARS1-RBD (**S1 Fig**), did not bind palm civet ACE2 efficiently. The ability of the PgCoV-RBD to efficiently bind human ACE2 but not palm civet ACE2, and the ability of the SARS2-RBD to bind pangolin ACE2, but not mouse, rat or palm civet ACE2, is consistent with a role for pangolin as a potential reservoir intermediate for SARS-CoV-2.

Finally we characterized in **Fig 1** five distinct horseshoe-bat ACE2 orthologs, namely those of *R. macrotis*, *R. pusillus*, *R. sinicus*, and two variants of *R. affinis* (clone 787 from Hubei, and clone 9479 from Yunnan) [27]. Despite the relatedness of these bats and the very similar expression levels of their ACE2 orthologs in HEK293T cells (**S1C Fig**), we observed a high level of variability in the ability of each RBD to bind these ACE2 orthologs. The SARS2-RBD bound *R. macrotis* and both *R. affinis* ACE2 orthologs, but did not bind *R. pusillus* or *R. sinicus* ACE2. The SARS1-RBD bound *R. affinis* ACE2 orthologs and strongly while bound R. sinicus ACE2 weakly. The RaTG13-RBD bound only one of two *R. affinis* orthologs, despite having been

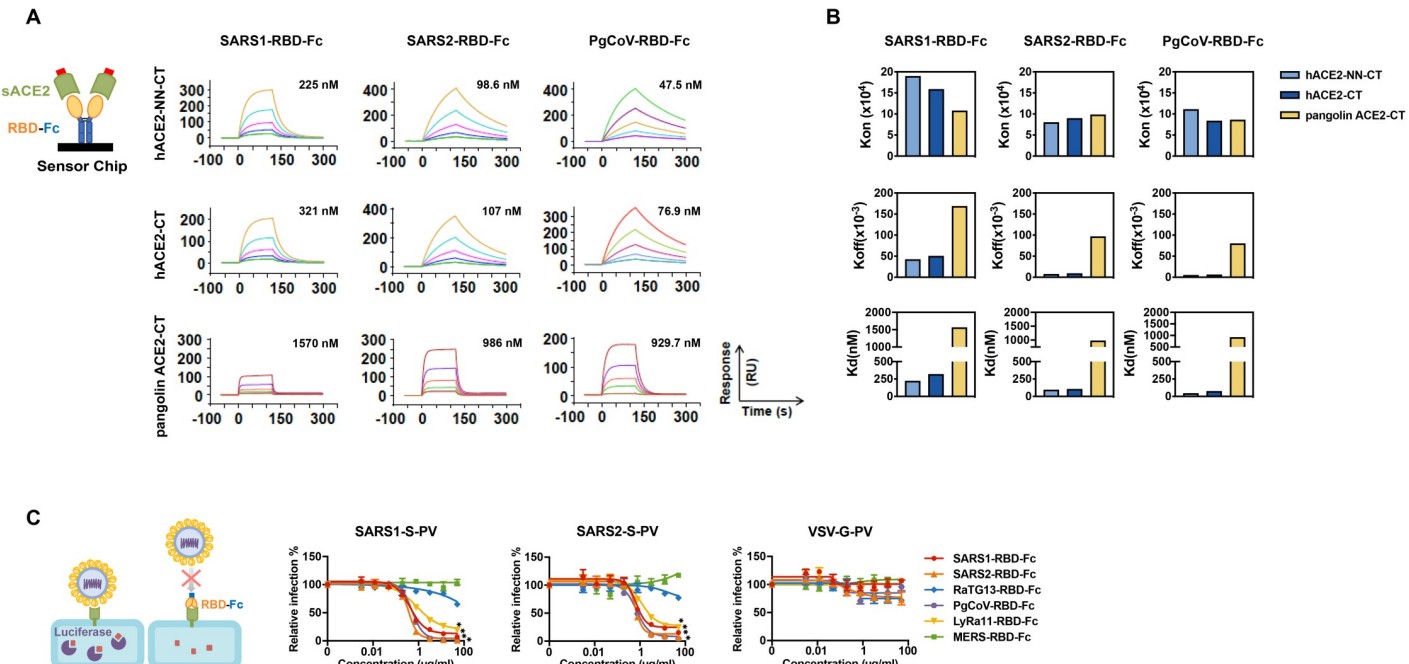

**Fig 2. Surface-plasmon resonance and pseudovirus inhibition studies of SARS1-RBD, SARS2-RBD, PgCoV-RBD, LyRa11-RBD.** (**A**) Biacore surface plasmon resonsance (SPR) studies were performed with the indicated RBD-Fc (yellow and blue) captured on the sensor chip. The soluble monomeric forms of the indicated ACE2 variants (green, with a four-amino-acid C-terminal tag in red) were injected for analysis as represented in the figure. Biacore X100 sensorgrams representing results from studies in which the indicated RBD-Fc variants were captured by an anti-human Fcγ antibody immobilized on a CM5 chip after instantaneous background depletion. Monomeric forms of tagged soluble human ACE2 (hACE2-CT), human ACE2 lacking its active-site histidines (hACE2-NN-CT), or pangolin ACE2-CT were injected at five concentrations, serially diluted by two-fold from the highest concentration (100 nM for human ACE2 variants). The experiment is representative of two with similar results. (**B**) The association rate constant ($k_{on}$), dissociation rate constant ($k_{off}$), and equilibrium dissociation constant ($K_d$) were calculated and plotted for comparison from the study shown in (A). (**C**) 293T-hACE2 cells were pre-incubated with the indicated concentrations of the indicated RBD-Fc variants, and subsequently infected with retroviral pseudoviruses (PV) pseudotyped with the S proteins of SARS-CoV-1 (SARS1-S-PV), SARS-CoV-2 (SAR2-S-PV), or with the G protein of vesicular stomatitis virus (VSV-G-PV), as diagrammed. Nonlinear fit for log(inhibitor) vs. response was calculated for each RBD. In both SARS1-S-PV and SARS2-S-PV panels, SARS-RBD-Fc, SARS2-RBD-Fc, PgCoV-RBD-Fc, LyRa11-RBD-Fc bottom plateaus CI did not overlap with values from VSV-G-PV. Error bars in indicate standard error of the mean (S.E.M) and are representative of results from at least two independent experiments. Asterisks indicate lack of 95% CI overlap of RBD-Fc constructs with respect to VSV-G group.

found in this species, and no other bat ACE2 assayed. None of the RBDs assayed associated with *R. pusillus* ACE2. Thus RBD from five SARS-like viruses, each presumably originating from horseshoe bats, bind horseshoe bat ACE2 orthologs with distinct patterns. The differential ability of these RBD to bind these ACE2 orthologs is reflected in a high level of amino-acid sequence variation in their RBD-binding regions (**Fig 3**). Collectively, these data suggest a high degree of selection pressure in horseshoe bats on this region of ACE2 from SARS-like viruses.

## Residues derived from horseshoe-bat ACE2 orthologs enhance neutralization of SARS2-CoV pseudoviruses

Uniquely among the horseshoe bat ACE2 orthologs assayed here, *R. affinis* (Yunnan) ACE2 bound all five RBD from SARS-like viruses. We noticed that the *R. affinis* (Yunnan) is composed solely of horseshoe-bat consensus residues in its ACE2 RBD-binding region (yellow in **Fig 3**). In contrast, *R. affinis* (Hubei) includes a non-consensus arginine at position 34, and *R. sinicus*, *R. macrotis*, and *R. pusillus* each have multiple unique residues in their ACE2 RBD-binding regions (**Figs 3 and S3**). This observation raised the possibility that SARS-like viruses are best adapted to ACE2 elements common among to the horseshoe bats where they circulate, and viral entry is somewhat impeded by unusual residues in their RBD-binding region. To test

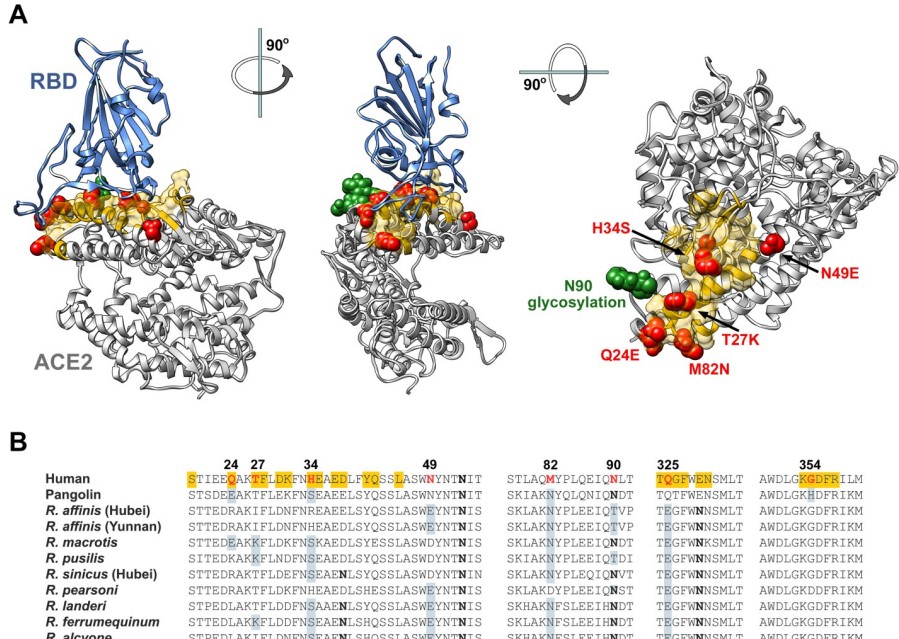

**Fig 3. Sequence comparison of essential ACE2 residues for SARS2-RBD binding in horseshoe bats and pangolin ACE2 orthologs. (A)** Human ACE2 (grey) is shown bound to the SARS-CoV-2 RBD (blue), based on PDB accession number 6M0J (Lan et al., 2020). Yellow indicates the ACE2 surface that is within 5.5 Å of the RBD. Red indicates residues that vary between human and horseshoe bat species. Specific mutations are indicated in the rightmost figure. A glycosylation at asparagine 90, also characterized, is indicated in green. Figure shows successive 90-degree rotations around the vertical and horizontal axes, respectively. The RBD is removed from the rightmost figure for clarity. **(B)** Sequence of human ACE2 is aligned with the sequence of the pangolin ACE2 as well as selected horseshoe-bat ACE2 orthologs. Residues, directly associating or proximal to, the SARS-CoV-2 RBD are shown. Yellow highlights a residue that directly contacts the RBD. Red indicates human residues that were altered to the indicated pangolin or bat residues shown in gray and characterized in the subsequent figures. Bold indicates a glycosylated asparagine.

this possibility, we replaced two non-consensus residues in *R. macrotis* ACE2 (K35 and E43) with horseshoe bat consensus amino acids (K35E, E43Q). Analogously, unique *R. pusillus* ACE2 residues at three positions were replaced bat consensus residues (K24E, N30D, D31K). Both sets of changes improved (*R. macrotis*) or rescued (*R. pusillus*) binding of SARS2-RBD and PgCoV-RBD to these ACE2 orthologs (S3B Fig). These changes also dramatically improved the efficiency of SARS2-S-PV entry in cells expressing *R. pusillus* (but not *R. macrotis*) ACE2 (S3C Fig). Similarly, modification of a unique histidine (H354) to glycine in the pangolin ACE2 ortholog improved binding of all four RBD that bound the unmodified ortholog (S3B Fig). These data show that consensus horseshoe-bat residues present in the RBD-binding region of *R. affinis* ACE2 facilitate association with five RBD from SARS-like viruses.

Based on these studies, we speculated that SARS-CoV-2 had not fully adapted to the use of the human receptor, and elements of horseshoe bat and perhaps pangolin ACE2 might enhance the affinity of human ACE2 for the SARS2-RBD. To test this hypothesis, we altered eight residues of an enzymatically inactive human ACE2 immunoadhesin (hACE2-NN-Fc, where "NN" indicates replacement of the active-site histidines with aspargines) to their analogues in one or more horseshoe-bat orthologs (red and green in Fig 3A). In addition to eight individual mutations based on this rationale (Q24E, T27K, H34S, N49D, N49E, M82N, N90D, Q325E), we further probed position 34 with residues present in other species (H34E, H34D, H34Y), and tested the impact of the pangolin ACE2 histidine at position 354 (G354H).

Human ACE2-NN-Fc and these twelve variants were produced in Expi293F cells (S4 Fig) and evaluated for their ability to neutralize SARS1-S-PV, SARS2-S-PV, RaTG13-S-PV and VSV-G-PV (Fig 4A–4C). Seven of these variants neutralized SARS2-S-PV more efficiently than did wild-type hACE2-NN-Fc (hACE2-NN-Fc Wt), namely: Q24E, T27K, H34S, H34D, N49E, M82N, and N90D. The ability of these hACE2-NN-Fc variants to neutralize SARS2-S-PV was precisely reflected in their binding affinities for monomeric SARS2-RBD, as determined by SPR (Figs 4D, 4E and S5). Notably, G354H, introducing an unusual pangolin residue, impaired SARS2-RBD association and SARS2-S-PV neutralization, and N90D, which removes a glycosylation at residue 90, markedly improved the SARS2-RBD binding by substantially increasing the SARS2-RBD on rate. All variants that neutralized SARS2-S-PV more efficiently bound SARS2-RBD with higher affinity than hACE2-NN-Fc Wt, and all variants that neutralized less efficiently bound with lower affinity. These patterns were consistent regardless of whether hACE2-NN-Fc (Figs 4D, 4E and S5) or SARS2-RBD-Fc (S6 Fig) was immobilized on the chip.

We then selected five best-performing bat-derived changes for additional characterization alone (Q24E, T27K, H34S, N49E, and N90D) or combined (5mut). These included two

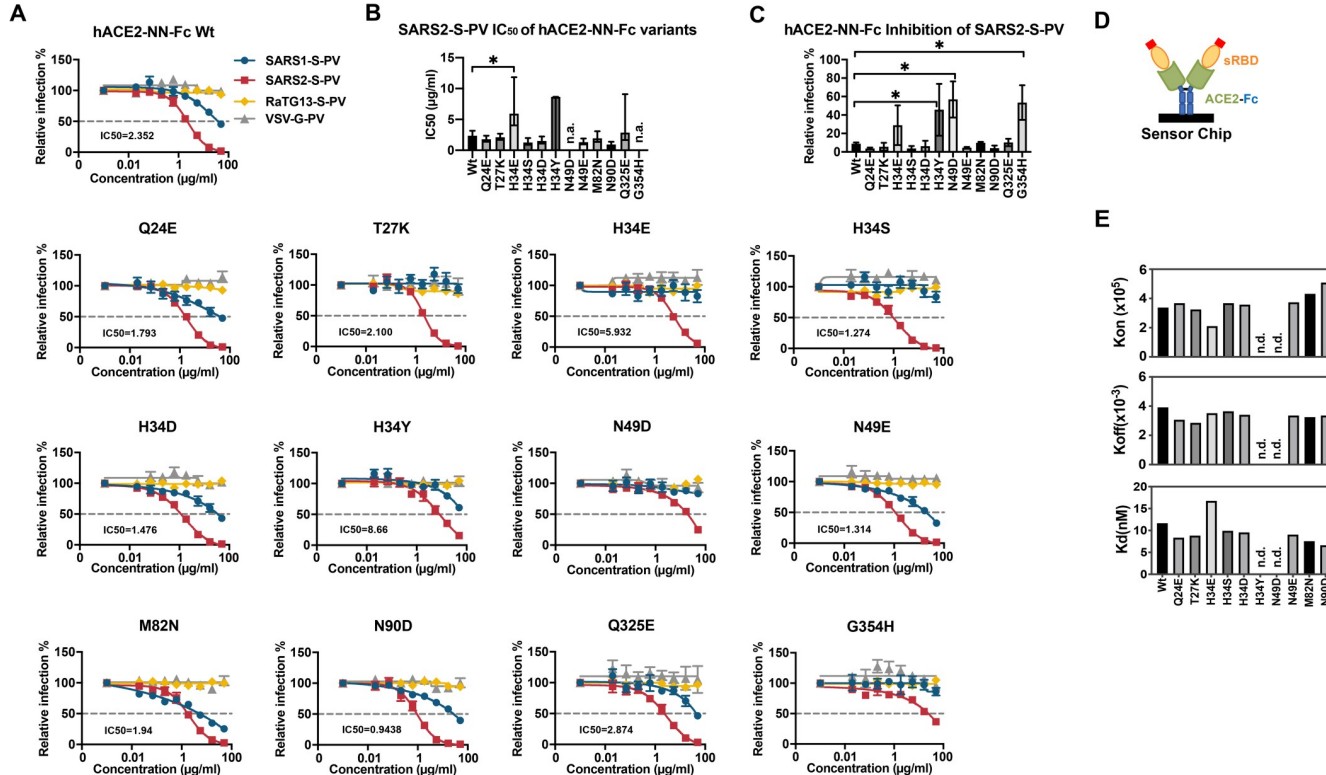

**Fig 4. Residues from horseshoe-bat ACE2 orthologs improved human ACE2-Fc binding to SARS2-RBD and neutralization of SARS2-S-PV.** (**A**) SARS1-S-PV (blue), SARS2-S-PV (red), RaTG13-S-PV (yellow) or VSV-G-PV (grey) were pre-incubated with the indicated concentrations of human ACE2-Fc lacking enzymatic activity (hACE2-NN-Fc) or hACE2-NN-Fc variants with the indicated mutations. 293T-hACE2 cells were incubated with these preincubated mixes and infection was analyzed 48 h post infection by measuring luciferase activity. Relative infection is calculated by dividing luciferase signal values to that in the absence of inhibitors. Error bars indicate S.E.M. and are representative of results from at least three independent experiments. Mean half-maximal inhibitory concentration ($IC_{50}$) to SARS2-S-PV is shown for each variant. (**B**) SARS2-S-PV $IC_{50}$ values for three studies shown in (A) are plotted. Error bars indicate the 95% confidence interval for $IC_{50}$ values observed for each hACE2-NN-Fc variant. n.a. indicates that no reliable $IC_{50}$ could be determined. (**C**) The relative percentage of SARS2-S-PV infection in the presence of 16.67 µg/ml indicated hACE2-NN-Fc variant are shown. Error bars indicate 95% confidence intervals and * indicates lack of overlap with Wt value. (**D**) A diagram is shown illustrating that hACE2-NN-Fc variants were captured and that soluble monomeric SARS2-RBD was injected for the studies showed in (E). (**E**) SPR analyses showing the $k_{on}$, $k_{off}$, and $K_d$ of SARS2-RBD to hACE2-Fc variants. n.d. is short for not detectable or no binding.

changes (Q24E; H34S) also present in pangolin ACE2. With the exception of T27K, each variant neutralized SARS2-S-PV more efficiently than hACE2-Fc Wt (**Figs 5A, 5B and S7A**), with 5mut neutralizing five-fold more efficiently (0.54 µg/ml, compared with 2.6 µg/ml for hACE2-Fc Wt). Consistent with these neutralization studies, the 5mut variant bound the SARS2-RBD with more than two-fold higher affinity (**Fig 5C and 5D**; 11.64 nM for hACE2-Fc Wt; 5.454 nM for 5mut when averaged over three experiments). Furthermore, the 5mut variant displayed increased neutralization activity to RaTG13-S-PV (**S7B Fig**) while there is almost no neutralization of this pseudovirus observed with wild-type hACE2-Fc or the variants bearing any single mutation (**Fig 4A**). Thus, individual mutations derived from horseshoe bats that enhance the neutralization potency of hACE2-NN-Fc can be combined to make an even more potent ACE2-Fc variant. However, hACE2-Fc 5mut could not neutralize SARS1-S-PV, likely a consequence of T27K and H34S, each of which impaired neutralization of SARS1-S-PV as individual mutations.

To investigate the neutralization activity of 5mut with cell lines more similar to the natural targets of SARS-CoV-2, we modified two human lung cell lines, NCI-H1975 and HBEC3-KT

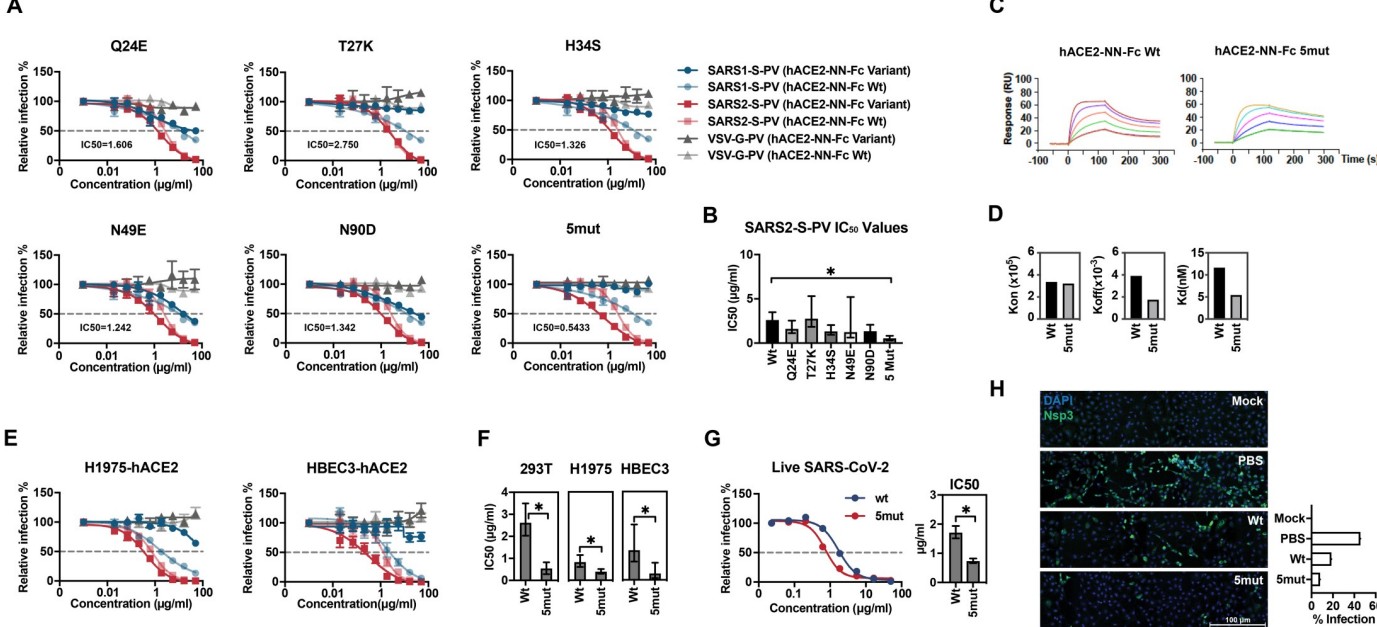

**Fig 5. A combination of five bat-derived residues improves SARS2-S-PV neutralization.** (**A**) Experiments similar to those shown Fig 4A except that the indicated ACE2-NN-Fc variant (dark red, blue and grey) is directly compared with wild-type human ACE2-NN-Fc (light red, blue, and grey) for their ability to neutralize SARS1-S-PV (light and dark blue), SARS2-S-PV (light and dark red), or VSV-G-PV (light and dark grey). Mean IC$_{50}$ values for SARS-CoV-2 of the indicated ACE2-NN-Fc variant over at least two independent studies are provided in each figure. Relative infection is calculated by dividing luciferase signal values to that in the absence of ACE2-Fc variants. Error bars indicate S.E.M. Neutralization data displays at least two independent experiments. (**B**) SARS2-S-PV IC$_{50}$ values shown in (A) are plotted. Error bars indicate 95% confidence intervals for IC$_{50}$ value observed for each hACE2-NN-Fc variant and * indicates lack of overlap with Wt value. (**C**) Sensorgrams of SPR analyses used to determine binding constants of the SARS2-RBD to wild-type hACE2-Fc (Wt) or hACE2-Fc with the five indicated mutations (5mut). (**D**) $k_{on}$, $k_{off}$ and K$_d$ of SARS2-RBD binding to hACE2-NN-Fc Wt or to hACE2-NN-Fc 5mut. (**E**) Neutralization potency of hACE2-Fc Wt and hACE2-Fc 5mut were compared with indicated two lung cell lines. Experiments similar to those shown Fig 5A, and color coded according to that figure, with darker colors indicating 5mut and lighter colors indicating ACE2-NN-Fc Wt. Neutralization data represents at least two independent experiments, with error bars indicating standard error of the mean (S.E.M.). (**F**) IC$_{50}$ values of hACE2-NN-Fc Wt and 5mut obtained from three different cell lines drawn from panels (A) and (E) were plotted, and error bars indicate 95% confidence intervals and * indicates lack of overlap with the Wt value. (**G**) Neutralization potency of hACE2-NN-Fc Wt and 5mut to SARS-CoV-2 infection were compared on Vero E6-hACE2 cells. Data represents at least two independent experiments, with error bars indicating standard error of the mean (S.E.M.). IC$_{50}$ values were plotted, and error bars indicate 95% confidence intervals and * indicates lack of overlap with the Wt value. (**H**) Vero E6-hACE2 cells were pre-incubated with PBS, 1 µg/ml hACE2-NN-Fc Wt or the 5mut variant, and subsequently mock-infected or infected with SARS-CoV-2 (MOI = 1). 24 h post infection, cells were fixed and stained with antibody against SARS-CoV-2 Nsp3 (green) and DAPI (blue). The scale bar indicates 100 µm. The infection percentage demonstrated with the bar chart was the percentage of Nsp3-positive cells from a pool of ~12,000 cells in each condition.

to stably express human ACE2 (H1975-hACE2 and HBEC3-hACE2, respectively). HBE-C3-hACE2 were selected to express relatively low levels of ACE2, whereas H1975-hACE2 expressed high levels and supported more SARS1- and SARS2- and RaTG13-S-PV transduction (**S8A and S8B Fig**). In both cases, 5mut neutralized SARS2-S-PV more efficiently than wild-type ACE2, but these differences were more dramatic with HBEC3-hACE2 when cellular ACE2 was limiting (**Fig 5E and 5F**). Specifically, the $IC_{50}$ on H1975-hACE2 differed between wild-type ACE2-NN-Fc and 5mut by 2.1-fold (0.392 µg/ml vs. 0.823 µg/ml), whereas they differed by HBEC3-hACE2 cells by 4.6-fold (0.293 µg/ml vs. 1.341 µg/ml) (**Fig 5F**). We also compared the neutralization potency of these two hACE2-Fc variants to live SARS-CoV-2 virus infection. Consistently, 5mut neutralized the SARS-CoV-2 virus more potently than the Wt protein with the $IC_{50}$ 0.734 µg/ml and 1.707 µg/ml respectively (**Fig 5G**). The infection was also visualized by staining infected or mock-infected cells with or without hACE2-Fc variants pre-incubation (**Fig 5H**). Twenty-four hours post infection, 45.72% cells, pre-treated with PBS, were infected while 18.60% cells were infected when pre-treated with 1 µg/ml hACE2-NN-Fc Wt. In contrast, only 8.11% cells were infected when cells were pre-treated with 1 µg/ml hACE2-NN-Fc 5mut. Thus the construct 5mut, bearing five residues derived from the consensus of horseshoe bat ACE2, neutralizes more efficiently than wild-type hACE-NN-Fc in multiple contexts.

## Discussion

Here we used several approaches to characterize the interaction between a number of ACE2 orthologs and the receptor-binding domains of SARS-CoV-1, SARS-CoV-2, and other SARS-like *betacoronaviruses*. As others have observed [4], the SARS2-RBD binds to human ACE2 with three-fold higher affinity than the SARS1-RBD (**Fig 2A and 2B**). Interestingly, the RBD from a SARS-CoV-2-like virus found in pangolins (PgCoV-RBD) bound human ACE2 as efficiently as the SARS2-RBD (**Figs 1 and 2A and 2B and S2B**), and the SARS2-RBD bound the pangolin ACE2 ortholog efficiently (**Figs 1 and S2B**). These data are consistent with, and may help solidify, the possibility that a pangolin served as a reservoir intermediate between horseshoe bats and humans.

Other data were more surprising. For example, the RBD from RaTG13, a SARS-CoV-2-like virus extracted from the horseshoe bat species *R. affinis*, detectably bound human, mouse and rat ACE2 orthologs but did not interact with four of five ACE2 orthologs isolated from various horseshoe bat species, including one of two ACE2 variants isolated from *R. affinis* itself (**Fig 1**). The ability of this RBD, as well as the PgCoV-RBD, to bind rodent ACE2 orthologs might be used to generate viruses better adapted to these rodents, perhaps providing an additional model of SARS-CoV-2 infection. The inability of the RaTG13-RBD to bind most horseshoe bat orthologs (**Fig 1**), and the high level of variability in horseshoe-bat ACE2 RBD-binding regions (**Fig 3**), may suggest that ongoing selection pressure from SARS-like viruses drives changes in these ACE2 regions. Consistent with this speculation, modification of an *R. pusillus* ACE2 ortholog to include three consensus horseshoe-bat sequences in its RBD-binding region rescued binding of the SARS1-, SARS2-, RaTG13-, and PgCoV-RBD (**S3 Fig**). Similarly, when two non-consensus residues of the *R. macrotis* ACE2 ortholog were reverted their horseshoe-bat consensus sequences, binding to the SARS2-RBD and PgCoV-RBD improved. Finally, when an unusual histidine in the pangolin ACE2 ortholog was reverted to a glycine present in all human and bat sequences, association with SARS2-RBD and the PgCoV-RBD also improved. These data suggest that most SARS-like viruses circulating among bats, and perhaps SARS-CoV-2 as well, will better associate with ACE2 variants bearing residues common to most horseshoe bats, rather than fully adapting to the bat ortholog from which these viruses

were most recently obtained. Thus the *R. affinis* ACE2 ortholog isolated in Yunnan is most similar to a consensus horseshoe bat sequence and efficiently binds all five *sarbecovirus* RBDs assayed (**Figs 1 and 3**). Indeed the consensus feature of its ACE2 may make *R. affinis* especially susceptible to any SARS-like viruses circulating among horseshoe bats.

We then asked whether this principle would extend to the human case. Specifically we hypothesized that consensus horseshoe-bat residues would increase the affinity of human ACE2 for the SARS-CoV-2 RBD. We accordingly generated a number of human ACE2-NN-Fc variants in which a human residue was replaced by that found in multiple horseshoe bat species (the 'NN' indicates mutations in the ACE2 active site, abrogating its endopeptidase activity). These changes included Q24E, T27K, H34S, N49D, M82N, and loss of a glycosylation site at N90 (**Fig 3**). Each of these changes modestly improved the ability of ACE2-NN-Fc to neutralize SARS2-S-PV (**Figs 4 and 5**). Increased neutralization corresponded to the increased ability of these ACE2-NN-Fc variants to bind the SARS2-RBD (**Figs 4 and S5 and S6**). A combination of five of these mutations generated an ACE2-NN-Fc variant that neutralized SARS-CoV-2 approximately 5-fold more efficiently than wild-type ACE2-NN-Fc, corresponding to a >2-fold enhancement of affinity for the SARS2-RBD, determined by surface plasmon resonance (**Fig 5**). Interestingly two of these mutations (T27K and H34S) impaired neutralization of SARS1-S-PV, implying they should be excluded from ACE2-Fc variants designed to anticipate future SARS epidemics. The remaining potency-enhancing mutations identified here could also be combined with D30E and the ACE2 collectrin domain, both also shown to enhance neutralization by ACE2-Fc [28]. Collectively these changes could improve the therapeutic utility of ACE2-Fc, currently under clinical evaluation, as a potential SARS-CoV-2 treatment.

## Materials and methods

### Plasmid

Protein sequence information used in this study was provided in **S1 Table**. Expression plasmids of coronavirus spike proteins or ACE2 proteins and variants were created by synthesizing fragments by Integrated DNA Technologies (IDT, Coralville, IA, USA), and ligating them into pCAGGS or pcDNA vector respectively using In-Fusion HD Cloning Kit (Takara Bio USA) according to manufacturer's instructions. ACE2-Fc variants were generated by the Quik-Change II site-directed mutagenesis protocol (Agilent). The plasmid expressing human DPP4 is a gift from Dr. Bart L. Haagmans.

### Cells and viruses

HEK293T (human embryonic kidney; ATCC) were maintained in growth media composed of Dulbecco's Modified Eagle Medium (DMEM, Life Technologies) supplemented with 2 mM Glutamine (Life Technologies), 1% non-essential amino acids (Life Technologies), 100 U/mL penicillin and 100 μg/mL streptomycin (Life Technologies), and 10% FBS (Sigma-Aldrich, St. Louis, MI, USA) at 37˚C in 5% $CO_2$. Expi293F cells (Thermo Fisher) were cultured in Expi293F expression medium (Thermo Fisher) at 37˚C in a shaking incubator at 125 rpm and 8% $CO_2$. NCI-H1975 cell line was kindly provided by Joseph Kissil (The Scripps Research Institute, Jupiter, FL), and were maintained in RPMI supplemented with 10% FBS and 100 U/mL penicillin and 100 μg/mL streptomycin. HBEC3-KT (ATCC) cell line was purchased from ATCC and maintained in airway epithelial cell basal medium supplemented with all the components included in bronchial epithelial cell growth kit (ATCC).

Cell lines expressing human ACE2 (hACE2) were created by transducing with vesicular stomatitis virus (VSV) G protein-pseudotyped murine leukemia viruses (MLV) containing

pQCXIP-myc-hACE2-c9 as previously described [24]. Briefly, HEK293T cells were co-transfected with three plasmids, pMLV-gag-pol, pCAGGS-VSV-G and pQCXIP-myc-hACE2-c9, and the medium was refreshed after overnight incubation of transfection mix. The supernatant with produced virus was harvested 72h post transfection and clarified by passing through 0.45 μm filter. Parental cells were transduced with generated MLV virus, and the hACE2 stably expressing cell lines were selected and maintained with medium containing puromycin (Sigma, 3 μg/ml for 293T and 1 μg/ml for H1975). hACE2 expression was confirmed by immunofluorescence staining using mouse monoclonal antibody against c-Myc antibody 9E10 (Thermo Fisher) and goat-anti-mouse IgG FITC (Jackson ImmunoResearch Laboratories, Inc) or SARS1-RBD-Fc or SARS2-RBD-Fc and goat-anti-human IgG APC (Jackson ImmunoResearch Laboratories, Inc). Vero E6-hACE2 was a gift from Jae U. Jung (Cleveland Clinic Lerner Research Center) and was maintained in DMEM containing 2 μg/ml puromycin.

MLV pseudotype viruses used for neutralization assay were produced by co-transfecting HEK293T cells with pMLV-gag-pol, pQC-Fluc and SARS-related spike proteins or VSV-G. SARS-CoV-2 live virus (strain 2019-nCoV/USA_WA1/2020) was kindly provided by Jae U. Jung and was propagated in Vero E6-hACE2 cells. All work relating to SARS-CoV-2 live virus was conducted in the BSL-3 facility of the Cleveland Clinic Florida Research and Innovation Center in accordance with institutional biosafety committee (IBC) regulations.

## Protein production and purification

Expi293F cells (Thermo-Fisher) were transiently transfected using FectoPRO (Polyplus) with plasmids encoding coronavirus RBDs or soluble ACE2 variants with a human-Fc fusion or a C-terminal C-tag (EPEA). After 5 days in shaker culture, media were collected and clarified of debris for 10 min at 1,500×g and filtered using 0.45-μm flasks (Nalgene). Proteins were purified using MabSelect Sure (GE Lifesciences) or CaptureSelect C-tagXL (Thermo-Fisher) columns according to the manufacturers' instructions. Eluates were buffer exchanged with PBS and concentrated using Amicon ultra filtration devices (Millipore Sigma) and stored at 4°C before use.

## Flow cytometry to test the binding of coronavirus RBD-Fc proteins to receptors

HEK293T cells were transfected with plasmids encoding ACE2 orthologs or human DPP4 using PEI 40K (Polysciences) according to manufacturer's instructions. 48h post transfection, transfected cells were detached and incubated with 5 μg/ml RBD-Fc, and the interaction was detected with goat-anti-human-Ig-APC (Jackson ImmunoResearch Laboratories, Inc). Expression of ACE2 orthologs were detected using mouse monoclonal antibody against c-Myc antibody 9E10 (Thermo Fisher) and Goat-anti-mouse FITC (Jackson ImmunoResearch Laboratories, Inc). Samples were analyzed by flow cytometry (BD Accuri C6 Flow Cytometry) and data was analyzed using FlowJo (FlowJo, LLC).

## Surface plasmon resonance studies

Kinetic and thermodynamic parameters for ACE2 binding to RBD-Fc or RBD binding to ACE2-Fc were measured on a Biacore X100 instrument as previously described [29]. Briefly, a CM5 sensor chip was immobilized with a mouse anti-human IgG CH2 mAb using reagents and instructions supplied with the Human Antibody Capture Kit (GE Healthcare) in order to capture RBD-Fc or ACE2-Fc. Monomeric ACE2 or monomeric RBD serially two-fold diluted in 1x HBS-EP+ running buffer were then injected respectively at five different concentrations with a replicate of the lowest concentration to confirm regeneration of the sensor chip.

Calculation of association ($k_{on}$) and dissociation ($k_{off}$) rate constants was based on a 1: 1 Langmuir binding model. The equilibrium dissociation constant ($K_d$) was calculated from $k_{off}/k_{on}$.

## Neutralization or inhibition studies of SARS1-S, SARS2-S or VSV-G pseudotyped MLV viruses

To investigate the inhibition of RBD-Fc proteins, 293T-hACE2 cells were pre-incubated with serially diluted proteins starting from 50 μg/ml. After one-hour incubation at 37˚C, pre-treated cells were inoculated with SARS1-S, SARS2-S or VSV-G pseudotyped MLV viruses at MOI = 1, and spun at 3000xg, 4˚C for 30 min. To compare the neutralization activity of ACE2 proteins, serially diluted proteins were pre-incubated with pseudotype viruses (MOI = 1) at 37˚C for one hour, and the mixes were added to hACE2 cell lines and spun at 3000×g, 4˚C for 30 min. Media was refreshed 2h after further incubation at 37˚C after the spin and firefly luciferase activity was measured (Britelight) 48 hours post-infection.

## Live virus neutralization assay

The neutralization activity of ACE2-NN-Fc variants against SARS-CoV-2 was determined by a microneutralization assay as described elsewhere with modifications [30]. Briefly, serially diluted proteins were pre-incubated with SARS-CoV-2 (MOI = 0.9) at 37˚C for one hour, and the protein-virus mixtures were added to Vero E6-hACE2 cells. The mixtures were removed from cells after one-hour adsorption at 37˚C and replaced with virus growth media (1×MEM with 2% FBS and 0.2% BSA) containing the serially diluted proteins. Cells were fixed with 4% formaldehyde at 24 hours post-infection, permeablized with 0.2% Triton X-100, and immunostained with anti-SARS-CoV-2 nucleocapsid (GeneTex 6H3) and HRP-linked anti-mouse IgG (CST #7076). Color was developed by using 1-Step Turbo TMB-ELISA Substrate Solution (Thermo Scientific) and the optical density measured at 652 nm.

## Immunofluorescence

Vero E6-hACE2 cells were grown on an 8-well Lab-Tek II CC2 chamber slide (Nunc) and infected with SARS-CoV-2 (MOI = 1) that was pre-incubated for 1 h with PBS, Wt, or 5mut ACE2-Fc (1 μg/ml). At 24 h post-infection, cells were fixed in 4% formaldehyde (in PBS) for 30 min, permeabilized with 0.2% Triton-X100 (in PBS) for 15 min, and blocked with 5% BSA (in PBS) for 1 h. Following incubation with anti-SARS-CoV-2 Nsp3 (1 μg/ml; GeneTex, GTX135589) in 5% BSA overnight, cells were washed and incubated with donkey anti-rabbit IgG Alexa Fluor 488 (2 μg/ml; Invitrogen) for 1 h, and with 4′,6-diamidino-2-phenylindole (DAPI) for 15 min. Cells were mounted in ProLong Gold antifade mountant (Invitrogen). Images were collected on a Leica Stellaris 8 point-scan confocal (Leica Microsystems), using a 20x/0.4 NA PL Fluostar L Dry objective (Leica Microsystems) and Leica hybrid detectors. A mosaic image for each well of the chamber slide was created using the tilescan and stitching/blending function of the LAS X software (version 4.1.0.23081). 10% overlap was used for tile registration, stitching, and blending. All individual tile images were captured at 512 x 512 pixels, with bi-directional X scanning at a speed of 600 Hz and 2x line averaging. The pinhole was set to 1 AU (106.2 μm). For the DAPI channel, sample was illuminated with a 405nm diode laser at 3.56% Leica Smart Intensity (measured 20 μW power at the objective focal point) and 44.9% Leica Smart Gain. For the Alexa Fluor 488 channel, sample was illuminated with a white light laser, filtered for 489nm excitation at 8.54% Leica Smart Intensity (measured 18 μW at the objective focal point) and 51.5% Leica Smart Gain. Images were analyzed in Fiji Image J (version 1.53h) using the Trainable Weka Segmentation plugin tool. The DAPI and Alexa

Fluor 488 channels were analyzed separately and the percentage of infected cells calculated by dividing the cell count in the Alexa Fluor 488 channel by that in the DAPI channel.

## Computational analysis

$IC_{50}$ analysis was performed after concentration was $log_{10}$ transformed, using default settings for log(inhibitor) vs. response Variable slope method (four-parameter model) in GraphPad Prism version 8.0.0, GraphPad Software, San Diego, California USA, www.graphpad.com. $IC_{50}$ values and 95% confidence intervals (CI) were calculated for complete dose-response curves. Due to incomplete dose-response curves, some $IC_{50}$ values were either labelled as unstable, or a best-fit value for $IC_{50}$ was calculated but a complete confidence interval was not, for negative controls or samples that had a similar profile to these controls. The samples with unstable $IC_{50}$ values and neither upper nor lower confidence intervals were omitted from the graphs. When negative control did not yield in confidence intervals for $IC_{50}$, bottom plateau was used to compare the curves.

## Supporting information

**S1 Fig. Sequence alignment and expression of coronavirus receptor binding domain (RBD) sequences and relative expression of ACE2 orthologs.** (**A**) Alignment of amino acid sequences of SARS or SARS-like coronavirus RBDs investigated in this study. Red indicated cysteine residues, bold indicates predicted N-glycosylation site, and grey indicates divergence from the SARS-CoV-2 RBD. The receptor-binding motif (RBM) region that directly contacts human ACE2 is indicated with a bracket. (**B**) Analysis of purified RBD-Fc proteins. Two micrograms of purified RBD-Fc proteins were analyzed on Novex 4–20% Tris-Glycine gradient gels under reducing (lower) and non-reducing (upper) conditions and stained with Bio-Safe Coomassie G-250 Stain. Position and sizes of the marker proteins are indicated. (**C**) HEK293T cells used in Fig 1 were transfected with control plasmid (pcDNA) or with plasmids expressing ACE2 orthologs and stained by anti-myc 9E10 antibody. ACE2 expression was determined by flow cytometry, and bars display the mean fluorescence intensity of stained cells from the three independent experiments as described in Fig 1.
(TIF)

**S2 Fig. Surface-plasmon resonance and neutralization studies with soluble human ACE2 variants.** (**A**) C-terminally tagged (CT) monomeric versions of soluble human ACE2, soluble human ACE2-NN and soluble pangolin ACE2 used in panel B were analyzed as in S1B Fig under non-reducing or reducing conditions as indicated. (**B**) A summary of numeric values obtained from surface-plasmon resonance studies shown in Fig 2. Soluble monomeric forms of the indicated ACE2 variants were injected for analysis with RBD-Fc variants were captured by an anti-human Fcγ antibody immobilized on a CM5 chip after instantaneous background depletion. Monomeric forms of human ACE2, human ACE2 with its active-site histidines mutated (ACE2-NN), or pangolin ACE2 were injected at five concentrations, serially diluted by two-fold from the highest concentration (100 nM for human ACE2 variants, 500 nM for nonhuman orthologs). (**C**) C-terminally tagged (CT) versions of soluble human ACE2 and soluble human ACE2-NN-Fc used in panel D were analyzed as in S1B Fig under non-reducing or reducing conditions as indicated. (**D**) The indicated concentrations of the indicated ACE2-Fc variants were incubated with retroviral pseudoviruses (PV) pseudotyped with the S proteins of SARS-CoV-1 (SARS1-S-PV), SARS-CoV-2 (SAR2-S-PV), or with the G protein of vesicular stomatitis virus (VSV-G-PV). Infection is normalized to that in the absence of inhibitors. Error bars indicate standard error of the mean (S.E.M), and are representative at least two

experiments with similar results. Asterisks indicate lack of 95% CI overlap of ACE2 constructs with respect to VSV-G group.
(TIF)

**S3 Fig. Hypothesized residues in ACE2 orthologs that are present in the real bat reservoir species.** (**A**) A sequence alignment similar to that in Fig 3B except that three variants in which consensus horseshoe bat residues are introduced, indicated in blue. (**B**) Studies similar to that in Fig 1 are shown, except that the indicated ACE2 orthologs are compared to variants mutated as shown in panel A. One-way ANOVA was used to test difference across RBDs within each ACE2 ortholog and upon low $p$ value, Dunnett's multiple comparison test was used to compare each RBD with SARS2-RBD-Fc (*indicates p<0.05). (**C**) Entry of SARS2-S-PV mediated by transiently expressed human, pangolin, or horse-shoe bat ACE2 orthologs. HEK293T cells were transfected with indicated ACE2 orthologs and then infected with SARS2-S-PV with VSV-G-PV and SARS2-S2-PV as positive and negative controls respectively. Luciferase signal was measured 48 h post transduction, and the data was normalized to the relative expression of ACE2 orthologs. Bars represent averages of at least two independent experiments. Error bars indicate standard error of the mean (S.E.M).
(TIF)

**S4 Fig. Expression of monomeric SARS2-RBD or ACE2-Fc variants used in Fig 4.** Purified monomeric SARS2-RBD or ACE2-Fc variants including any of the single mutation indicated were analyzed under reducing (left) and non-reducing (right) conditions as described in S1 Fig.
(TIF)

**S5 Fig. SPR analyses for the binding of monomeric SARS2-RBD to human ACE2-Fc variants, as summarized in Fig 4E.** Analysis similar to Fig 2 except that ACE2-Fc variants were captured on the chip and monomeric SARS2-RBD was injected in. (**A**) Biacore X100 sensorgrams obtained. (**B**) Numerical values of $k_{on}$, $k_{off}$ and $K_d$ from the analysis.
(TIF)

**S6 Fig. Comparative SPR analyses for the binding of monomeric human ACE2 variants to SARS2-RBD-Fc captured on the chip, as summarized in Fig 5D.** (**A**) Purified monomeric ACE2 variants were analyzed on 4–20% Tris-Glycine gradient gel under reducing (upper) and non-reducing (lower) conditions as in S1 Fig. (**B**) The Biacore X100 sensorgrams of the indicated monomeric ACE2 various bound to immobilized SARS2-RBD-Fc are shown. $K_{on}$, $K_{off}$, and $K_d$ from this analysis that are presented in a table (**C**) and plotted in (**D**). (**E**) A representation of the experiment in (B) indicating that the RBD-Fc was captured and that soluble monomeric ACE2 variants were captured.
(TIF)

**S7 Fig. Expression of ACE2-Fc variants used in Fig 5 and neutralization activities of ACE2-Fc variants to RaTG13-S-PV.** (**A**) Two micrograms of ACE2-Fc variants including either single mutation or all five mutations were analyzed on Novex 4–20% Tris-Glycine gradient gel under reducing (lower) and non-reducing (upper) conditions. (**B**) Human ACE2-Fc 5mut demonstrated weak neutralization activity to RaTG13-S-PV. Experiments similar to those shown in Fig 4A except that the neutralization of ACE2-Fc variants to RaTG13-S-PV was directly compared. Relative infection is calculated by dividing luciferase signal values to that in the absence of ACE2-Fc variants. Error bars indicate standard error of the mean (S.E.M). Neutralization data displays at least two independent experiments.
(TIF)

**S8 Fig. Surface staining and infection of cell lines with and without hACE2.** (**A**) HEK293T cells, NCI-H1975 cells, and HBEC3 cells with and without human ACE2 (hACE2) transduction were stained with 5 ug/ml SARS1-RBD-Fc or SARS2-RBD-Fc, and the binding was detected by goat-anti-human IgG-APC. (**B**) HEK293T cells with and without hACE2 expression in (A) were transduced with serial-diluted SARS2-S-PV, and the entry efficiency was measured with luciferase signal. (**C**) Similar experiments were performed with NCI-H1975 cells and HBEC3 cells with and without hACE2 expression, and assayed for transduction with the indicated pseudoviruses.
(TIF)

**S1 Table. Information for the expression plasmids used in this study.** Protein name, accession number and plasmid vector for each expression plasmid used in this study is provided.
(TIF)

## Acknowledgments

We appreciate technical assistance by John Heddleston from the Imaging Core of The Cleveland Clinic Florida Research and Innovation Center (FRIC) for the IF analysis.

## Author Contributions

**Conceptualization:** Huihui Mou, Michael Farzan.

**Funding acquisition:** Michael Farzan.

**Investigation:** Huihui Mou, Brian D. Quinlan, Haiyong Peng, Guanqun Liu, Yan Guo, Shoujiao Peng, Lizhou Zhang, Lindsey B. DeVaux, Zhi Xiang Voo.

**Methodology:** Huihui Mou, Haiyong Peng, Guanqun Liu, Lizhou Zhang, Meredith E. Davis-Gardner, Matthew R. Gardner.

**Resources:** Matthew R. Gardner, Charles C. Bailey, Michael D. Alpert.

**Software:** Huihui Mou, Gogce Crynen.

**Supervision:** Christoph Rader, Michaela U. Gack, Hyeryun Choe, Michael Farzan.

**Visualization:** Huihui Mou, Haiyong Peng, Guanqun Liu, Lizhou Zhang, Michael Farzan.

**Writing – original draft:** Huihui Mou, Michael Farzan.

**Writing – review & editing:** Huihui Mou, Haiyong Peng, Guanqun Liu, Yan Guo, Lizhou Zhang, Meredith E. Davis-Gardner, Matthew R. Gardner, Michael Farzan.

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
