## [Decision Letter · Decision Letter 0]

4 Jan 2021

Dear Dr. Farzan,

Thank you very much for submitting your manuscript "Mutations derived from horseshoe bat ACE2 orthologs enhance ACE2-Fc neutralization of SARS-CoV-2" for consideration at PLOS Pathogens. As with all papers reviewed by the journal, your manuscript was reviewed by members of the editorial board and by several independent reviewers. In light of the reviews (below this email), we would like to invite the resubmission of a significantly-revised version that takes into account the reviewers' comments.

The reviewers mostly agreed that the study has merits in distinguishing between the various horseshoe bat ACE2 orthologs.  However, I agree with reviewer #2 that at this point in the epidemic, confirmation of any antiviral activity of ACE2 based reagents must be made in the context of live virus infection, the field is now mature enough that not only live viruses should be used but potentially different isolates that now have mutations in the RBD domains. At the minimum, the key data to confirm that ACE2-Fc orthologs from cognate horseshoe bats are 'potent neutralizers' of SARS-CoV-2 must be performed with relevant live SARS-CoV-2 on susceptible human cells or cell lines. In the absence of live virus data, the claims of this paper does not rise to the standard of advice we require at PLoS Pathogens. 

We cannot make any decision about publication until we have seen the revised manuscript and your response to the reviewers' comments. Your revised manuscript is also likely to be sent to reviewers for further evaluation.

Sincerely,

Benhur Lee

Section Editor

PLOS Pathogens

Benhur Lee

Section Editor

PLOS Pathogens

Kasturi Haldar

Editor-in-Chief

PLOS Pathogens

orcid.org/0000-0001-5065-158X

Michael Malim

Editor-in-Chief

PLOS Pathogens

orcid.org/0000-0002-7699-2064

Reviewer's Responses to Questions

**Part I - Summary**

Reviewer #1: The manuscript by Mou, Quinlan, Peng and colleagues describes an analysis of the binding affinities of the Receptor Binding Domain (RBD) from several SARS-like coronaviruses with several orthologues of the receptor protein ACE2. Binding affinity is analyzed by different techniques: 1) binding of soluble RBD to cells expressing ACE2 measured by flow cytometry, 2) surface-plasmon resonance (SPR) using both, immobilized RBD versus soluble ACE2 and immobilized ACE2 versus soluble RBD and 3) soluble ACE2 mediated neutralization of infection by pseudo-typed viruses in cell lines expressing ACE2. In general there is good correlation between the results from the different techniques and with previous published results. With the exception of a couple of typos, the manuscript, figures and supporting information are quite clear.

The results show striking differences between the binding affinities of different combinations of RBD-ACE2 which could have implications in the elucidation of the origin and evolution of the SARS-like coronaviruses. Interestingly, the ACE2 orthologues from several bats species of the same genus (suspected to be the original reservoir of the SARS-CoVs ancestors) show very different binding patterns and the authors briefly speculate that it could be indicative of selection pressure on the bat host to limit infection susceptibility.

The results described in this manuscript have 2 potential applications for SARS-CoV2 research/treatment: 1) the description of RBD from animal SARS-like viruses that bind more efficiently to the ACE2 orthologue of laboratory animals (mouse, rat) could contribute to the development of animal models for SARS-CoV2. 2) Introduction of bat-like amino acid changes in the soluble human ACE2 increase binding affinity and neutralization, with a potential application in therapy.

Reviewer #2: This study by Mou et al describes a detailed investigation of the impact of key mutations of various bat ACE2 molecules on their ability to interact with the RBD of different SARSr-CoVs. The key findings are: a) even for horseshoe bat ACE2 molecules from the same species, there can be important functional difference in the context of RBD binding; b) SARS-CoV and SARS-CoV-2 most likely originated from different horseshoe bat species; c) the SARS-CoV-2 RBD/human ACE2 interaction can still be improved indicating it is not at optimal status yet; d) the human ACE2-Fc incorporating several consensus horseshoe bat ACE2 residues increased its potency to neutralize SARS-CoV-2 PV and can potentially replace potent mAbs as SARS-CoV-2 therapeutics.

**Part II – Major Issues: Key Experiments Required for Acceptance**

Reviewer #1: The experimental approach is adequate and the conclusions are well supported by the results, without over interpretation of the data. I don’t see the need for additional experiments.

Reviewer #2: Overall, the study is well conducted and some of the findings are of significance to the ongoing research in the context of the “special” relationship between horseshoe bats and SARSr-CoVs. However, the following issues/aspects need to be addressed to improve the current study.

1) Despite the recognised importance of ACE2 in SARSr-CoV infection and emergence, there is no conclusive data to suggest that ACE2 is the only/major entry receptor for all SARSr-CoVs in bats. In fact, the first group of SARS-like-CoVs reported in 2005 (by Li et al) contain Spike proteins which failed to bind ACE2 molecules from the bat species the sequences were derived from.

2) In the same context, the authors need to discuss the limitation of the current study (in fact all published studies on bat ACE2 molecules), i.e., all of the studies were exclusively carried out in the human cell environment. We therefore can not be certain that the findings will be the same if the more relevant bat cells are used in such studies.

3) Despite the usefulness and obvious advantage of no need for BSL3 containment for the PV system used exclusively in this study, I am disappointed not seeing any validation at all using live virus for the current study. There have been reported discrepancies between PV and live virus for SARSr-CoV investigation. I would strongly recommend validation for a few key findings. Especially for the “potent neutralization of SARS-CoV-2” aspect in the context of promoting the mutated human ACE2-Fc as potential therapeutics.

4) The authors claimed the mutated human ACE2-Fc has equal or better neutralizing potency as some of the best mAbs. I would like to see side-by-side comparison data on live virus to support such a claim.

**Part III – Minor Issues: Editorial and Data Presentation Modifications**

Reviewer #1: The manuscript has a couple of minor errors that should be corrected:

Line 34 (abstract) “closely related SARS-CoV-2,” should say “closely related to SARS-CoV-2,”

Line 51 (author summary) “by interacting the cellular receptor” should be “by interacting with the cellular receptor”

Line 52 (author summary) “is mediated the viral spike” should be “is mediated by the viral spike”

Line 54 (author summary) “with the binding of affinity of” should be “with the binding affinity of”

Line 486 (Figure 2 legend): “were performed the indicated RBD-Fc” should be “were performed with the indicated RBD-Fc”

Reviewer #2: Lines 34-35 in Abstract: Please replace “RaTG13 …. has been isolated from …” to “RaTG13 …. has been detected from …”. Same for the rest of the paper, please avoid the use of “isolate or isolated” for genome sequences detected in bats. These should be reserved for those with live virus isolation only. This is generating a lot of media confusion: unfortunately, not many journalists can tell the BIG difference between isolation of live virus and detection of genetic sequences in bats. We should hence try our best to reduce such confusion in our scientific publications.

PLOS authors have the option to publish the peer review history of their article (what does this mean?). If published, this will include your full peer review and any attached files.

Reviewer #1: No

Reviewer #2: No
---

## [Editor Report · Decision Letter 1]

24 Mar 2021

Dear Dr. Farzan,

We are pleased to inform you that your manuscript 'Mutations derived from horseshoe bat ACE2 orthologs enhance ACE2-Fc neutralization of SARS-CoV-2' has been provisionally accepted for publication in PLOS Pathogens.

Best regards,

Benhur Lee

Section Editor

PLOS Pathogens

Benhur Lee

Section Editor

PLOS Pathogens

Kasturi Haldar

Editor-in-Chief

PLOS Pathogens

orcid.org/0000-0001-5065-158X

Michael Malim

Editor-in-Chief

PLOS Pathogens

orcid.org/0000-0002-7699-2064

Reviewer Comments (if any, and for reference):  The authors have made a good faith effort in addressing the major comments of the reviewers.

---

## [Editor Report · Acceptance letter]

4 Apr 2021

Dear Dr. Farzan,

We are delighted to inform you that your manuscript, "Mutations derived from horseshoe bat ACE2 orthologs enhance ACE2-Fc neutralization of SARS-CoV-2," has been formally accepted for publication in PLOS Pathogens.

Best regards,

Kasturi Haldar

Editor-in-Chief

PLOS Pathogens

orcid.org/0000-0001-5065-158X

Michael Malim

Editor-in-Chief

PLOS Pathogens

orcid.org/0000-0002-7699-2064